# A Systematic Review of Uremic Toxin Concentrations and Cardiovascular Risk Markers in Pediatric Chronic Kidney Disease

**DOI:** 10.3390/toxins16080345

**Published:** 2024-08-08

**Authors:** Heshini Dalpathadu, Aly Muhammad Salim, Andrew Wade, Steven C. Greenway

**Affiliations:** 1Department of Medical Sciences, Cumming School of Medicine, University of Calgary, 3330 Hospital Drive NW, Calgary, AB T2N 4N1, Canada; heshini.dalpathadu@ucalgary.ca; 2Department of Cardiac Sciences and Libin Cardiovascular Institute, Cumming School of Medicine, University of Calgary, 3330 Hospital Drive NW, Calgary, AB T2N 4N1, Canada; 3Department of Pediatrics and Alberta Children’s Hospital Research Institute, Cumming School of Medicine, University of Calgary, Alberta Children’s Hospital, 28 Oki Drive NW, Calgary, AB T3B 6A8, Canada; awwade@me.com; 4Department of Neuroscience and Hotchkiss Brain Institute, Cumming School of Medicine, University of Calgary, 3330 Hospital Drive NW, Calgary, AB T2N 4N1, Canada; alymuhammad.salim@ucalgary.ca; 5Department of Pediatric Nephrology, Cumming School of Medicine, University of Calgary, 3330 Hospital Drive NW, Calgary, AB T2N 4N1, Canada; 6Department of Biochemistry & Molecular Biology, Cumming School of Medicine, University of Calgary, 3330 Hospital Drive NW, Calgary, AB T2N 4N1, Canada

**Keywords:** cardiorenal syndrome, uremic toxin, uremic solute, uremic cardiomyopathy, cardiac toxicity, chronic kidney disease, pediatric, systematic review

## Abstract

Chronic kidney disease (CKD) can lead to cardiac dysfunction in a condition known as cardiorenal syndrome (CRS). It is postulated that the accumulation of uremic toxins in the bloodstream, as a consequence of declining kidney function, may contribute to these adverse cardiac effects. While CRS in adults has been extensively studied, there is a significant knowledge gap with pediatric patients. Uremic toxin levels in children remain inadequately characterized and quantified compared to adults. This review aims to systematically evaluate the association between uremic toxin concentrations and cardiac changes in pediatric CRS and to examine the impact of different dialysis modalities, specifically hemodialysis and peritoneal dialysis, on uremic toxin clearance and cardiovascular parameters. To address this, we conducted a systematic literature search of PubMed, following PRISMA guidelines. We used the terms “uremic toxins” and “cardiorenal syndrome” with variations in syntax to search for studies discussing the relationship between uremic toxin levels in CKD, the subsequent impact on cardiac parameters, and the emergence of cardiac dysfunction. Full-text articles written in English, conducted on humans aged from birth to 18 years, and published until December 2021 were included. A comprehensive literature search yielded six studies, and their risk of bias was assessed using JBI Critical Appraisal Checklists. Our systematic review is registered on PROSPERO, number CRD42023460072. This synthesis intends to provide an understanding of the role of uremic toxins in pediatric CRS. The findings reveal that pediatric patients with end-stage CKD on dialysis exhibit elevated uremic toxin levels, which are significantly associated with cardiovascular disease parameters. Additionally, the severity of CKD correlated with higher uremic toxin levels. No conclusive evidence was found to support the superiority of either hemodialysis or peritoneal dialysis in terms of uremic toxin clearance or cardiovascular outcomes. More pediatric-specific standardized and longitudinal studies are needed to develop targeted treatments and improve clinical outcomes and the quality of life for affected children.

## 1. Introduction

Cardiorenal syndrome (CRS) encompasses a collection of disorders involving the heart and the kidneys where acute or chronic dysfunction of one organ leads to acute or chronic dysfunction in the other organ [1,2]. This crosstalk between the two organs may be direct or indirect and is postulated to involve a complex feedback loop (Figure 1) [3]. CRS is phenotyped as cardiorenal or renocardiac based on the order of organ involvement and further classified into five subtypes based on disease severity (acute or chronic) and sequential organ dysfunction: type 1 (acute heart failure leading to acute kidney injury), type 2 (chronic heart failure leading to progressive chronic kidney disease (CKD)), type 3 (acute kidney injury leading to acute heart failure), type 4 (CKD resulting in chronic heart failure), and type 5 (systemic conditions like autoimmune diseases and sepsis affecting both heart and kidneys) [2,4].

In pediatric patients, several key mechanisms contribute to the pathogenesis of CRS, including hemodynamic alterations such as fluid overload and renal hypoperfusion, neurohormonal activation of the renin-angiotensin-aldosterone system and sympathetic nervous system, chronic inflammation, oxidative stress, and the accumulation of uremic toxins [5]. Uremic toxins, a heterogeneous collection of metabolites normally filtered from the blood and excreted by healthy kidneys, accumulate in the bloodstream as renal function declines [6,7]. As bioactive compounds, these toxins disrupt normal biological processes through cytotoxicity, disrupted cellular signaling, induction of oxidative stress, and systemic inflammation [7,8]. Uremic toxins are defined based on their ability to show measurable biological activity, such as their impact on enzyme function, hormone responses, receptor activity, and cellular toxicity in experimental settings [9].

Uremic toxins are categorized into three broad classes based on their molecular weight, physicochemical properties, binding properties, and behavior during dialysis [8]: small water-soluble uremic toxins, middle molecule uremic toxins, and protein-bound uremic toxins [10]. This review delves into several uremic toxins with proposed or documented pathogenic impacts on cardiovascular (CV) health. The small water-soluble uremic toxins ADMA (asymmetric dimethylarginine) and SDMA (symmetric dimethylarginine) promote endothelial dysfunction and the subsequent development of atherosclerosis and other CV clinical manifestations [11]. ADMA is also an endogenous inhibitor of nitric oxide synthase, leading to impaired nitric oxide signaling and, therefore, vascular dysfunction [6]. β2-microglobulin (β2M), a middle molecule uremic toxin, is associated with vascular stiffness, inflammation, and bone and joint degradation in CKD patients [8]. The protein-bound uremic toxins IS (indoxyl sulfate) and pCS (p-cresyl sulfate) induce oxidative stress and endothelial dysfunction, serving as markers and predictors of CV disease in CKD patients [12,13,14]. Other protein-bound uremic toxins, such as indoleacetic acid (IAA) and hippuric acid (HA), contribute to inflammation, oxidative stress, renal fibrosis, and endothelial dysfunction [6,15].

Children with CKD face compounded CV risks due to the combined impact of uremic toxins and traditional risk factors [5]. The majority of CV deaths in children with CKD are due to cardiac arrest, followed by arrhythmia and cardiomyopathy [5]. The developing nature of pediatric organs and unique pediatric-specific conditions make the CV system more susceptible to uremic toxin effects [5]. Despite the critical need, most research on uremic toxins has focused on adult CKD patients, and limited research exists on the presence and impact of uremic toxins in children [16]. Findings in adults may not directly translate to children due to physiological differences (e.g., larger body water volumes in children), the absence of pre-existing cardiovascular diseases (CVDs) and traditional risk factors (e.g., diabetes mellitus, coronary artery disease, and peripheral artery disease) in children, distinct intestinal microbiota, the potential influence of growth and puberty on maturational processes, and the unique etiologies of CKD in children (i.e., congenital anomalies of the kidney and urinary tract and hereditary renal diseases) compared to adults (i.e., glomerulopathies) [16,17]. Discrepant levels of toxins between children and adults may also indicate different origins of the toxins and variations in elimination [8].

Our primary research goal is to explore the relationship between uremic toxins and cardiac parameter changes in pediatric CRS and assess the impact of different dialysis modalities on the clearance of these toxins and subsequent CV outcomes. This systematic review aims to consolidate existing research on uremic toxin concentrations and related CV risk markers in children with CRS type 4, assess the correlations, and highlight the differences in toxin accumulation and the consequences between pediatric and adult populations.

Addressing these knowledge gaps is crucial for developing pediatric-specific treatment guidelines and improving the quality of life for children with CRS. Future research should focus on patient-centered outcomes such as growth, development, and quality of life, and explore innovative approaches such as personalized medicine, dietary interventions, and gut microbiota modulation to manage uremic toxin levels. By integrating these novel perspectives, we aim to enhance the understanding and management of CRS in pediatric patients.

## 2. Results

A PRISMA flow chart [18] summarizes the study selection (Figure 2). A total of 92 studies were initially identified. After screening titles and abstracts, 76 studies were excluded. Following the assessment of 16 full-text articles, six studies were selected for this systematic review (Table 1). These eligible articles varied in study design, participant demographics, and the types of uremic toxins and cardiac parameters assessed (Table 1 and Appendix A). The study designs were either cross-sectional (three studies), cohort studies (two studies), or case-control studies (one study) (Table 1). Upon quality assessment, it was found that all studies demonstrated low or moderate risk of bias (Figure 3). Uremic toxins discussed among the six eligible studies were classified according to class as small water-soluble uremic toxins, middle molecule uremic toxins, and protein-bound uremic toxins (Figure 4). Any associations mentioned between uremic toxin concentrations and CV parameter changes are summarized in Appendix A.

### 2.1. Patient Characteristics and eGFR Levels

To provide a comprehensive overview of the patient populations in the included studies, we examined the patient characteristics and estimated glomerular filtration rate (eGFR) levels (Table 1). Crespo-Salgado et al. focused on established dialysis patients, including those on peritoneal dialysis (PD) or hemodialysis (HD) [19]. Although the study did not specify exact eGFR values of the patient population, the healthy control group consisted of children with a median eGFR of 118 [85; 139] mL/min/1.73 m^2^, using the modified Schwartz equation [19].

Holle et al. included pre-dialysis CKD patients with eGFR values categorized across CKD stages [20]. The reported eGFR values were as follows: an overall mean of 28.1 ± 10.3 mL/min/1.73 m^2^, CKD stage 3a at 51.4 ± 4.1 mL/min/1.73 m^2^, stage 3b at 36.0 ± 4.1 mL/min/1.73 m^2^, stage 4 at 22.5 ± 4.1 mL/min/1.73 m^2^, and stage 5 at 13.3 ± 1.4 mL/min/1.73 m^2^ [20]. Similarly, in their subsequent study, Holle et al. reported on pre-dialysis CKD patients with an overall mean eGFR of 26.6 ± 11.2 mL/min/1.73 m^2^ [21].

Özdemir et al. included established dialysis patients (HD and PD) but did not specify the eGFR levels [22]. The control group in this study comprised healthy children; however, their specific eGFR values were not provided [22].

The first study by Snauwaert et al. examined pre-dialysis CKD patients with detailed eGFR values across CKD stages [23]. The authors reported median eGFR values of 74 [67; 103] mL/min/1.73 m^2^ for CKD stages 1–2, 47 [35; 55] mL/min/1.73 m^2^ for stage 3, 20 [18; 26] mL/min/1.73 m^2^ for stage 4, and 11 [9; 13] mL/min/1.73 m^2^ for stage 5, with an overall median eGFR of 48 [24; 71] mL/min/1.73 m^2^ [23]. The study included a healthy control group with a median eGFR value of 137 [119; 159] mL/min/1.73 m^2^ [23].

The subsequent paper by Snauwaert et al. focused on pre-dialysis CKD patients in stages 4–5, reporting a median eGFR of 17 [11; 23] mL/min/1.73 m^2^ [24]. The healthy control group in this study also had a median eGFR of 137 [119; 159] mL/min/1.73 m^2^ [24].

Understanding the patient characteristics and eGFR levels is crucial as it helps contextualize the variations in uremic toxin levels among different pediatric CKD stages. With this foundation, we now explore the uremic toxin levels in healthy children, providing a baseline reference for further analysis.

### 2.2. Uremic Toxin Levels in Healthy Children

In examining the uremic toxin levels reported in the included studies, we found that specific values for healthy pediatric controls were provided in detail by Snauwaert et al. [23]. This study presented reference values for a range of uremic toxins in healthy children stratified by age and sex [23]. The reported concentrations for healthy controls were 0.39 ± 0.16 mg/dL for creatinine, 0.64 ± 0.08 µmol/L for SDMA, 0.67 ± 0.11 µmol/L for ADMA, 1.71 ± 0.43 µg/mL for complement factor D (CfD), 1.74 ± 0.34 µg/mL for β2M, 0.006 ± 0.005 mg/dL for p-cresyl glucuronide (pCG), 0.044 ± 0.037 mg/dL for HA, 0.023 ± 0.010 mg/dL for IAA, 0.056 ± 0.025 mg/dL for IS, 0.244 ± 0.179 mg/dL for pCS, and 0.010 ± 0.012 mg/dL for 3-carboxy-4-methyl-5-propyl-furanpropionic acid (CMPF) [23]. These values serve as an important comparative baseline for evaluating the uremic toxin burden in pediatric patients with CKD [23]. The other studies either did not include healthy control groups or did not report specific values for uremic toxin levels in their control populations.

Based on the updated Schwartz equation, the median eGFR in healthy children was 137 [119; 159] mL/min/1.73 m^2^, classified as normal under the Kidney Disease Improving Global Outcomes guidelines, indicating the absence of CKD unless other kidney disease markers are present [23].

Establishing these baseline values allows us to better understand the extent of uremic toxin accumulation in pediatric CKD patients. By contrasting these levels with those found in CKD patients, we can delineate the impact of declining kidney function on uremic toxin concentrations and their potential role in CRS.

### 2.3. Pediatric Uremic Toxin Concentrations

We have compiled and summarized the concentrations of uremic toxins, categorizing them by class based on the quantitative values reported across studies, the assays used for toxin measurement, and the timing of sample collection (Appendix A).

Crespo-Salgado et al. discovered that patients with end-stage kidney disease (ESKD) undergoing PD or HD had significantly increased levels of the toxins IS and pCS compared to healthy controls and those who had received a kidney transplant [19,25]. They also discovered that serum IS and pCS levels were comparable in healthy controls and recipients of kidney transplants [19]. The study highlighted the differential accumulation of these toxins based on dialysis modality, with PD showing better clearance than HD. However, a study by Holle et al., where they examined pediatric patients with CKD stages 1–5, both supported and contradicted these findings [21]. In this second study, a significant association was found between serum IS levels and CKD progression in pediatric patients, independent of other known risk factors such as baseline eGFR, proteinuria, and blood pressure [21]. However, no correlation was observed between pCS and CKD progression [21]. The highest hazard ratio associated with high IS levels was noted in underweight and possibly malnourished patients [21]. Therapeutics, such as the oral absorbent AST-120 and intensified dialysis modalities, were unsuccessful in lowering serum IS levels [21].

In a 2018 study, Snauwaert et al. established reference values for a range of uremic toxins in healthy children and children with non-dialysis CKD stages 1–5 [23]. The authors showed that concentrations of SDMA, CfD, β2M, IS, pCS, IAA, CMPF, and HA were higher in children with CKD (all stages) compared to healthy children [23]. However, when the CKD group was subdivided into stages, pCS, pCG, and IAA concentrations were higher in those with more severe disease (stages 3–5), and ADMA and HA were higher than controls only in patients with stage 5 disease [23]. This observation is consistent with previous studies showing that the accumulation of uremic toxins is related to the severity of the renal disease [6]. Of the reported reference values for the same set of uremic toxins in a healthy pediatric population, they found that some toxin concentrations were similar to those of healthy adults while others were decreased [23]. In healthy children, the concentrations of SDMA, IS, pCS, IAA, and β2M were comparable to those found in healthy adults, whereas the concentrations of CfD, CMPF, HA, and pCG were lower [23].

In a later study, Snauwaert et al. measured the concentrations of six protein-bound uremic toxins (pCG, HA, IAA, IS, pCS, and CMPF) in a large cohort of children on HD (*n* = 170) compared to children with non-dialysis CKD stages 4–5 (*n* = 24) and healthy controls (*n* = 50) [24]. They found that children with CKD on maintenance HD had higher plasma levels of pCG, HA, IAA, IS, and β2M compared to children with CKD stages 4–5 who were not on any type of dialysis, but the levels of uric acid, pCS, and CMPF were similar in both groups [24]. This suggests that HD affects certain toxins differently, and this observation has potential implications for CKD treatment strategies. In the HD group, pCG, HA, and IS levels were positively correlated with age and negatively correlated with residual urine volume [24]. Residual kidney function (RKF), defined as the ability of the diseased kidneys to eliminate water and uremic toxins, plays a crucial role in the health outcomes of pediatric dialysis patients [24]. Compared to anuric (severe reduction or absence of urine production) children on HD, RKF contributes to better clearance of protein-bound uremic toxins in non-anuric children on HD [24]. This enhanced clearance is associated with improved growth, better volume, mineral and electrolyte control, and decreased CV morbidity [24]. Studies have consistently shown that pediatric patients with preserved RKF experience fewer CV complications and have better overall health outcomes than those without RKF. Furthermore, Snauwaert et al. discovered higher levels of the unbound forms of the protein-bound uremic toxins pCG, HA, IAA, and IS in children on HD compared to children with CKD who were not on dialysis [24]. Factors contributing to this modified binding affinity may include lower albumin levels resulting from an increased rate of protein catabolism and/or albumin loss and a lower binding affinity to albumin due to post-translational modifications such as oxidation, carbamylation, nitrosylation, glycation, and acetylation [24].

Several factors other than renal excretion may influence the lack of clearance of some uremic toxins. These factors include tubular function, the activity of organic anion transporters OAT-1 and OAT-3, consumption of high-protein and high-phosphate diets, changes in intestinal transit time, alterations in gut microbial metabolism, and the compromised activity of toxin-degrading enzymes involved in metabolic processes [26]. In addition to these factors, we also propose the following possible reasons for variations in uremic toxin clearance: impaired liver metabolism, medication interactions, dehydration or overhydration affecting kidney function, inflammation, infections, changes in hormones that influence kidney function, and genetic variability.

### 2.4. Relationship between Uremic Toxin Levels and Cardiovascular Risk Markers

Holle et al. correlated serum levels of IS and pCS with vascular measurements, patient demographics, and clinical parameters in 609 pediatric patients with CKD [20]. They discovered that IS and pCS levels were inversely correlated with eGFR in children [20]. Furthermore, their analysis showed that IS was significantly correlated with baseline carotid intima-media thickness, a reliable surrogate marker and a potent predictor of future CVD [27,28]. In addition, Holle et al. observed that the change in pulse wave velocity was significantly associated with IS levels [20,29]. Pulse wave velocity is currently considered the best method for measuring arterial stiffness and is also an independent prognostic marker for CVDs [29]. Finally, the authors observed that pCS levels were positively associated with age, serum albumin, and non-Mediterranean residency and negatively associated with glomerular disease [20]. The observations from this study suggest that higher levels of IS may be associated with the development of CVD in children, as indicated by thicker carotid artery walls and increased arterial stiffness.

Özdemir et al. examined structural and functional CV alterations in 20 children with CKD on dialysis [22]. The authors compared the CV parameters of the patient group (those undergoing HD and PD) with those of the control group [22]. Four out of 10 patients on HD and two out of 10 patients on PD presented with significantly worse functional CV parameters (augmentation index and pulse wave velocity) and structural CV parameters (carotid intima-media thickness and left ventricular mass index) compared to the control group [22]. The mean values of these CV parameters were higher in this subset of six patients with CVD (the four HD patients and two PD patients) than in the controls [22]. The remaining patients in both the HD and PD groups did not show any CV malfunctions [22].

To assess uremic toxin clearance, Özdemir et al. analyzed at least two toxins from each of the three classes in blood samples collected 30 min before dialysis (D0) and 2 h after dialysis (D1) [22]. The rate of change was calculated using the formula: (D0 − D1)/D0 [22]. The four HD patients with CV involvement had significantly lower clearance rates of β2M (a middle molecule uremic toxin) and homocysteine (a protein-bound uremic toxin) compared to those undergoing HD without CV involvement [22]. In contrast, there was no significant difference in uremic toxin levels between patients receiving PD with and without CV complications [22]. There was no difference in the clearance of small water-soluble uremic toxins between patients on either type of dialysis with or without cardiac dysfunction [22]. The authors concluded that PD was more efficient than HD in the clearance of middle molecule and protein-bound uremic toxins, which may impact CV outcomes [22].

In summary, the findings from these studies (Table 2) suggest that the improvement in cardiac function post-dialysis in pediatric CKD patients may be closely linked to the clearance of specific uremic toxins and classes. Key toxins implicated include IS, pCS, β2M, and homocysteine. These toxins are associated with endothelial dysfunction, increased arterial stiffness, and left ventricular hypertrophy (LVH), all of which contribute to CVD [6]. While not all studies provide direct measurements of uremic toxin levels before and after dialysis, the evidence from Özdemir et al. supports the hypothesis that improved cardiac function may be due to the increased clearance of specific uremic toxins, particularly those classified as middle molecule and protein-bound uremic toxins.

Our review highlights the importance of targeted dialysis protocols in managing uremic toxins and calls for future research to focus on the longitudinal tracking of uremic toxin levels pre- and post-dialysis, alongside CV outcomes, to provide stronger evidence and better guide clinical practice.

### 2.5. Distinctions between Dialysis and Non-Dialysis Patients

CV system involvement can begin early in pediatric patients with CKD, even before the initiation of dialysis [22]. Early manifestations include LVH and increased arterial stiffness [19,22]. These CV alterations progress with the severity of kidney disease [5]. Pediatric patients with CKD but not yet on dialysis often exhibit premature CV system involvement, highlighting the need for early intervention and continuous monitoring to prevent progression to severe CVD once dialysis is initiated [22]. Therefore, prioritizing CV care in pediatric CKD patients is crucial to mitigate long-term CV risks [22].

Both traditional and non-traditional CV risk factors are prevalent in pediatric CKD patients, regardless of dialysis status [22]. Traditional risk factors such as hypertension, dyslipidemia, and anemia are common in both groups, but their impact is more pronounced in dialysis patients due to the additional uremic burden [22]. Non-traditional factors, including inflammation, oxidative stress, and the accumulation of middle and large molecular weight uremic toxins, significantly contribute to CV morbidity and mortality in pediatric patients on dialysis [22]. The clearance efficiency of small water-soluble uremic toxins is generally satisfactory in both dialysis and non-dialysis patients; however, CV outcomes are not solely dependent on this toxin class [22]. The role of middle molecule and large protein-bound uremic toxins, which are not as efficiently cleared by conventional dialysis methods, is increasingly recognized as a critical factor in cardiac dysfunction [22]. Studies have shown that inadequate clearance of these toxins is associated with increased arterial stiffness, vascular calcification, and a higher incidence of CV events [22].

Pediatric patients on dialysis exhibit significant differences in uremic toxin clearance and CV parameters compared to their non-dialysis counterparts, with variations observed between HD and PD modalities [22]. While some studies suggest a high incidence of LVH among patients undergoing HD compared to those receiving continuous ambulatory PD [22], conclusive evidence favoring one dialysis modality over the other regarding CV risk profiles remains elusive. In their study of pediatric dialysis patients, Özdemir et al. found that 40% of patients undergoing HD exhibited LVH, compared to 20% in those receiving PD [22]. Their findings also indicate significantly greater clearance of β2M (a middle molecule uremic toxin) and homocysteine (a protein-bound uremic toxin) in HD patients without CVD compared to HD patients with CVD [22]. However, it’s important to note that improved clearance of specific toxins such as β2M and homocysteine does not necessarily imply or translate to improved CVD outcomes. Moreover, the clearance of the toxins mentioned above does not guarantee the clearance of all uremic toxins in their respective classes. Current literature indicates that HD inadequately removes protein-bound uremic toxins such as IS and pCS, which are consistently associated with adverse CV events [30]. Therefore, the effectiveness of HD in removing CV-linked uremic toxins and mitigating CVD outcomes remains uncertain.

As evidenced by previous research, CKD patients are exposed to high levels of uremic toxins daily, which increases their risk of CVD [8]. However, in contrast to HD patients, PD patients in the Özdemir et al. study did not show significant differences in uremic toxin levels between those with and without CVD [22]. This lack of significant difference should not be interpreted as an indication that PD provides superior toxin clearance than HD. Based solely on this information, it is not possible to draw conclusions about the superiority of PD over HD in toxin clearance, as Özdemir et al. did not address potential confounding factors or include initial toxin levels in their study [22]. Interpreting PD as potentially offering greater toxin clearance is nuanced and requires consideration of factors such as patient selection, dialysis regimen, and overall clinical management. For instance, a study conducted in adult ESKD patients demonstrated that PD was more effective in improving the clearance of molecular substances such as urea nitrogen and whole parathyroid hormone compared to HD [31]. However, no longitudinal studies directly comparing PD versus HD, with measurements of toxin levels pre- and post-dialysis, have been conducted in the pediatric population to date. These complexities highlight the importance of dialysis modality selection in mitigating CV risk among pediatric patients, suggesting that each modality may offer distinct advantages in reducing CV complications through targeted removal of specific uremic toxins and classes. Therefore, it is essential to approach the topic with caution and acknowledge the limitations of current evidence. 

## 3. Discussion

Although the existing literature on pediatric CRS is sparse, we found compelling evidence that supports an association between levels of uremic toxins and CV risk factors. Studies, such as those by Crespo-Salgado et al. and Holle et al., demonstrated elevated uremic toxin levels in pediatric patients on dialysis compared to healthy controls and those with non-dialysis CKD [19,20]. The impact of specific dialysis modalities on reducing uremic toxin levels and potentially improving cardiac function was also evident, particularly in the study by Özdemir et al. [22]. These findings align with previous adult studies but provide novel insights specific to the pediatric population. Some conflicting observations in the included literature may be related to differences between study populations.

The association between elevated levels of gut-derived uremic toxins and adverse cardiac risk markers is important, as these toxins, in conjunction with phosphate minerals, contribute to bone disorders by inducing parathyroid hormone resistance [20]. Parathyroid hormone resistance can lead to secondary hyperparathyroidism, a common CKD complication, which results in vascular and valvular calcification and the development of atherosclerosis [32]. Targeted therapeutics, such as phosphate binders and vitamin D receptor agonists, can aid in treating CKD-induced secondary hyperparathyroidism by reducing serum phosphate levels and increasing osteopontin levels, which regulate bone remodeling [33,34,35]. Managing gut-derived uremic toxins can assist in mitigating bone disorders and reducing CV risk in CKD patients.

There are inconsistent data on pCS levels in pediatric CKD [19,21]. Moreover, the positive association between pCS levels and non-Mediterranean residency, as observed by Holle et al., could be due to geographically based food consumption patterns [20]. The traditional Mediterranean diet consists of a relatively low amount of dietary protein, which could contribute to the lower production of protein-bound uremic toxins by the intestinal microbiota, suggesting diet as a feasible therapeutic target for effectively managing and treating CRS [36,37].

Despite obesity being a CV risk factor in adults with CKD, underweight children presented with the highest hazard ratio for the relative contribution of high IS levels to renal survival [21]. Pediatric CKD patients tend to be underweight due to poor nutritional intake, which can cause alterations in their intestinal microbiota, thus affecting uremic toxin production [38]. Therefore, targeting the gut microbiome may present a promising avenue for therapeutic intervention in pediatric CRS. Furthermore, this underscores the necessity for pediatric-specific research and treatment approaches, as adult findings cannot be directly applied to children due to differences in nutritional status, microbiota composition, and disease etiology.

Our analysis indicates that the choice of dialysis modality can impact uremic toxin clearance and CV outcomes in pediatric patients. PD was previously believed to be better at removing solutes than HD, a claim supported by Özdemir and colleagues’ finding that PD was more efficient than HD at clearing middle molecule and protein-bound uremic toxins [22,39]. Despite these findings, studies have shown that plasma levels of uremic toxins are not consistently different between PD and HD patients [39]. Additionally, although both PD and HD may allow for the rapid or short-term removal of uremic toxins, these reductions are temporary and not sustained, resulting in limited long-term impact on uremic toxin levels [40,41,42]. Despite there being no clear advantage between PD and HD in terms of uremic toxin removal, PD provides gentle ultrafiltration [43]. In contrast, ultrafiltration through HD is associated with myocardial stunning, which can result in persistent left ventricular dysfunction, leading to systolic dysfunction and subsequent heart failure [43]. Improvements in HD, such as increased frequency and extended duration, have not improved the clearance or the control of protein-bound uremic toxins, resulting in limited effectiveness in decelerating and preventing CV damage caused by this class of toxins [39,40,41,42,44]. The strong binding affinity between protein-bound uremic toxins, such as IS and pCS, and circulating proteins, such as albumin, limits their removal during HD and could explain why dialysis modalities are less efficient in removing these uremic toxins when compared with kidney transplantation [19,44,45]. Modifications such as hemodiafiltration and high-flux membranes have shown promise in removing middle molecule uremic toxins more effectively by increasing convection and pore size [46,47,48,49]. However, the impact of hemodiafiltration and high-flux membranes on the removal of protein-bound uremic toxins is limited, with studies indicating that these techniques do not significantly reduce plasma concentrations of protein-bound uremic toxins compared to conventional HD [44,45,46,49]. These findings suggest that HD may not be an effective strategy for removing protein-bound uremic toxins in pediatric CKD patients, and, thus, it would not adequately address the cardiac toxicity caused by elevated levels of these toxins. This conclusion highlights the urgent need for alternative therapeutic approaches to manage protein-bound uremic toxins and mitigate CV risks in this vulnerable population.

The choice between HD and PD for removing uremic toxins depends on several factors: the class and structure of uremic toxins, patient-specific factors such as age, comorbidities, residual kidney function, and vascular access, and treatment goals, including fluid management, electrolyte balance, and patient comfort. Ultimately, the decision between HD and PD should be personalized based on the patient’s clinical status, preferences, and the specific uremic toxins targeted for removal. However, the inability to use HD in infants and very young or small children will limit these types of comparative studies.

Although studies investigating uremic toxin levels in transplanted versus non-transplanted CKD patients are limited, available data show that serum IS and pCS levels significantly decrease after kidney transplantation [50]. Possible explanations for the normalization of uremic toxin levels following transplant include improved kidney function, intestinal microbiota alterations, and the use and interaction of drugs such as immunosuppressants and antibiotics [50].

The prevalence of traditional and non-traditional CV risk factors in both dialysis and non-dialysis pediatric CKD patients indicates a complex interplay of influences on CV health. Traditional CV risk factors are exacerbated and further compounded by the additional burden of non-traditional CV risk factors, such as the accumulation of middle and large molecular weight uremic toxins in dialysis patients. Moreover, the recognition that CV outcomes are not solely dependent on the clearance of small molecular weight uremic toxins but are critically influenced by middle and large molecular weight uremic toxins necessitates a more comprehensive approach to managing these patients [22].

The findings from our synthesis highlight distinctions between pediatric patients on dialysis and those not yet initiated on dialysis. The early involvement of the CV system in pediatric CKD patients underscores the necessity for proactive monitoring and intervention strategies. Even prior to dialysis initiation, children with CKD exhibit CV alterations such as LVH and increased arterial stiffness, which progress with the severity of kidney disease [22]. This early involvement calls for regular CV assessments in pediatric CKD patients to identify and mitigate potential risks before they escalate into severe CVD once dialysis begins.

A major limitation of our review is the scarcity of research on pediatric CRS, leading to limited studies meeting our criteria and the inclusion of studies with considerable heterogeneity. The majority of CKD research has been performed in adults, and the identity and relative concentrations of uremic toxins in children are poorly defined [16]. Typically, children with CKD present with different CKD etiology, disease mechanisms, disease-modifying factors, and even differing epigenetic changes compared to adults [51]. The influence of pubertal growth and sexual maturation may also be a factor in the progression of pediatric CKD and may be a cause of age-specific differences in toxin concentrations [52]. Studies often do not evaluate factors that could influence uremic toxin levels, such as hormonal changes during puberty, diet, obesity, genetic influences, and medications impacting the gut microbiota [21,53,54]. Another limitation was the inconsistency in uremic toxin measurement techniques across studies, with methods such as high-performance liquid chromatography and enzyme-linked immunosorbent assays being commonly used. This variability in assay techniques is important because it can lead to differences in sensitivity, specificity, and accuracy, thereby affecting the comparability and reliability of the results across studies. The timing of sample collection (pre-dialysis, post-dialysis, or on non-dialysis days) also varied, highlighting the need for standardized measurement protocols. Finally, the majority of reviewed studies focused on correlation and, therefore, definitive causal relationships between toxin levels and CVD cannot be established. Without establishing causality, developing targeted therapeutic interventions and fully understanding the mechanisms by which uremic toxins contribute to CVD in pediatric CKD patients remains challenging.

## 4. Clinical Implications

The distinct CV profiles of pediatric patients on and off dialysis highlight the need for tailored clinical approaches. For patients on dialysis, regular monitoring of uremic toxin levels and CV parameters is essential. Interventions such as personalized dialysis regimens, dietary modifications, and pharmacological treatments targeting specific uremic toxins should be explored to mitigate CV risk. Clinicians should consider the advantages of PD, compared to HD, in reducing CV risk by effectively clearing middle and large molecular weight uremic toxins [22]. Additionally, the preservation of RKF should be a primary goal in the management of pediatric dialysis patients, with efforts directed toward interventions that support and preserve RKF [24].

Given the significant correlation between IS levels and CV risk markers, clinicians should consider incorporating regular assessments of these toxins into routine care for pediatric CKD patients. Regarding the dialysis modality debate, clinicians should consider other factors such as patient preference, lifestyle and specific clinical needs and conditions when choosing between HD and PD rather than significant uremic toxin clearance or CV outcomes. Regular cardiac monitoring is imperative for the early detection and timely intervention of CVD in pediatric CKD patients, whether they are on dialysis or not. This proactive approach can help manage and mitigate the progression of CVD, improving long-term health outcomes.

## 5. Future Research Directions

Future research should focus on longitudinal studies to better understand the long-term CV outcomes associated with different dialysis modalities in pediatric patients. Studies exploring the mechanisms underlying the differential clearance of uremic toxins between HD and PD and determining if one dialysis modality is superior in removing specific classes of uremic toxins based on their molecular size and binding properties can provide valuable insights for optimizing dialysis protocols.

Research should also delve into the development of new dialysis technologies or therapies that enhance the clearance of middle and large molecular weight uremic toxins, potentially offering better protection against CVDs. Furthermore, studies examining the role of non-traditional CV risk factors, such as inflammation and oxidative stress, in pediatric CKD patients can lead to the identification of mechanistic pathways and, therefore, novel therapeutic targets.

The role of nutritional interventions in managing uremic toxin levels and improving CV outcomes in pediatric CKD patients should also be explored. Studies investigating the impact of specific dietary components, such as low-protein diets or supplementation with specific nutrients, on uremic toxin levels and CV health could provide practical insights for clinical management. Additionally, future research should evaluate the potential of gut microbiota-targeted therapies. Given the significant role of the gut microbiome in uremic toxin production, interventions such as probiotics, prebiotics, or fecal microbiota transplantation could offer novel avenues for reducing toxin levels and improving CV outcomes.

Finally, there is a need for well-controlled pediatric studies to establish pediatric-specific reference ranges for uremic toxins and to understand their unique impact on the pediatric population. Such research will help inform the development of childhood-specific treatment guidelines and strategies, optimize renal replacement therapies, and potentially use uremic toxin levels as biomarkers for identifying CV risk and disease.

## 6. Conclusions

In our review of the limited pediatric literature, significant associations were found between uremic toxin concentrations and key CVD parameters, underscoring the potential role of uremic toxins in CV complications in children. These connections were especially notable in patients undergoing dialysis compared to healthy controls and post-kidney transplant patients. The limited research on CRS in children reveals potential future therapeutic targets, including enhancing dialysis techniques (through high-flux membranes or by increasing the removal of the free solute fraction to drive dissociation of the protein-toxin complex), targeting the microbiota, and modifying diet.

The pediatric population offers a compelling alternative for the study of CRS due to the absence of pre-existing confounding comorbidities and CVDs. This review supports the need for well-controlled pediatric studies and the establishment of pediatric-specific reference ranges for uremic toxins. The improved understanding and management of uremic toxins could reduce the CV burden in this vulnerable population, ultimately improving their quality of life, long-term prognosis, and clinical outcomes.

## 7. Methods

This systematic review was conducted according to the Preferred Reporting Items for Systematic Reviews and Meta-Analyses (PRISMA) recommendations [18] and registered with PROSPERO (CRD42023460072).

### 7.1. Article Selection Criteria

Our inclusion criteria encompassed original research articles written in English and published in peer-reviewed journals. We specifically included studies involving patients with CRS type 4 (CKD resulting in chronic heart failure). The age range of the patient population was restricted to between birth and 18 years of age. Conversely, we excluded abstracts, review articles, commentaries, books, book chapters, editorials, conference proceedings, protocol papers, and other non-original research material. Additionally, we excluded non-human studies, research involving adult subjects, articles on CRS types 1, 2, 3, and 5, studies where the primary disease was acute kidney injury, and studies in which cardiac dysfunction preceded renal impairment in the sequence of organ involvement.

### 7.2. Search Strategy

A PubMed search was performed to identify eligible studies published before December 2021. We used the following search string: (uremic toxins OR uraemic toxins) AND (cardiorenal syndrome OR cardio renal syndrome OR cardio-renal syndrome OR cardiorenal syndrome type 4 OR cardiorenal syndrome type IV OR cardio renal syndrome type 4 OR cardio renal syndrome type IV OR cardio-renal syndrome type 4 OR cardio-renal syndrome type IV). This comprehensive search strategy ensured the inclusion of studies relevant to our research question, focusing on CRS type 4 based on the chronicity of the condition and the order of the disease process where dysfunction originates in the kidneys.

### 7.3. Data Collection and Analysis

Two reviewers (H.D. and A.M.S.) independently evaluated the titles and abstracts of all eligible studies. The same reviewers then screened the complete texts of the selected articles. When an eligible paper was identified, the reviewers screened its reference list and also evaluated articles that had cited it and articles flagged as similar by PubMed for additional potentially relevant articles. If consensus was not reached, the article was assessed by additional reviewers (S.C.G. and A.W.). The reviewers performed data extraction on all eligible articles (Table 1). Designated extraction forms were completed independently, with any discrepancies resolved through discussion. The authors used an electronic data extraction form to include the following variables and details: first author, published country, period of patient recruitment, type of study, total number of participants per study, gender, mean age, uremic toxins studied, correlations between uremic toxins and cardiac effects, and main findings. In terms of quantitative values of uremic toxins, we extracted means and standard deviations or medians and interquartile ranges of these toxins in CKD patients (Appendix A). The normal reference ranges for these toxins in children were obtained from Snauwaert et al. [23].

### 7.4. Quality Assessment and Risk of Bias

We used the Joanna Briggs Institute (JBI) Critical Appraisal Tool to assess the quality of cross-sectional studies, cohort studies, and case-control studies. Two reviewers (H.D. and A.M.S.) independently evaluated the quality of each study, specifically, methodological rigor, including but not limited to sample size, control for confounding variables, and measurement reliability. Discrepancies were resolved through discussion. Based on the checklist, a JBI score, presented as a percentage, was calculated for each study. A study with a JBI score ranging from 20% to 49% was considered to have a high risk of bias, while a score of 50% to 79% indicated a moderate risk of bias, and a score of 80% to 100% suggested a low risk of bias.

## Figures and Tables

**Figure 1 toxins-16-00345-f001:**
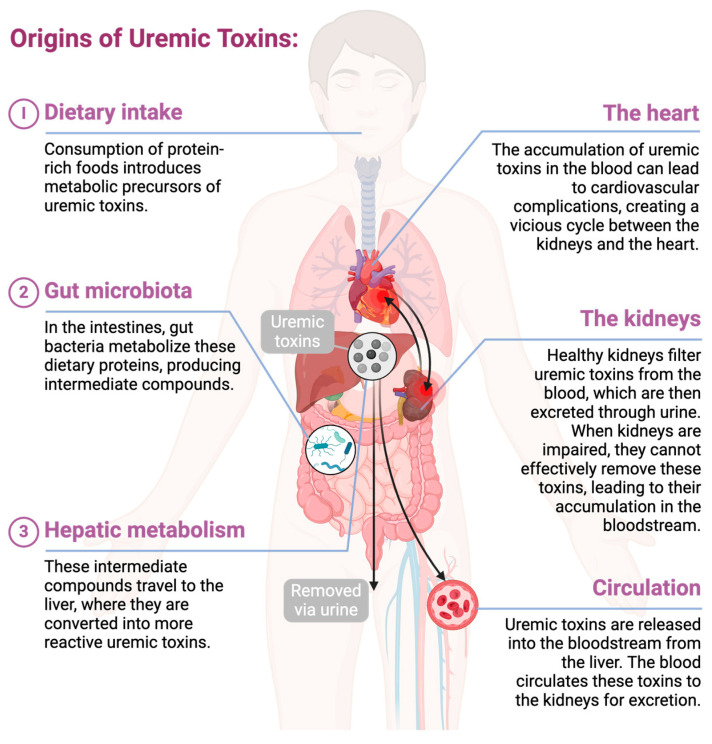
The journey of uremic toxins from their generation to elimination. Uremic toxins are metabolic by-products or waste substances originating from dietary intake, gastrointestinal tract, and liver. These toxins enter the bloodstream and are subsequently transported to the kidneys for filtration and removal from the body in the urine. In chronic kidney disease, the effective elimination of these toxins is compromised. The accumulated uremic toxins can cause cardiovascular complications through myocardial toxicity and systemic inflammatory responses. This cardiac dysfunction can reduce renal perfusion and enhance fluid retention due to decreased cardiac output, further exacerbating cardiac load. The resulting renal dysfunction, characterized by inadequate filtration and fluid imbalance, perpetuates a feedback loop, worsening the progression of cardiorenal syndrome. Created with BioRender.

**Figure 2 toxins-16-00345-f002:**
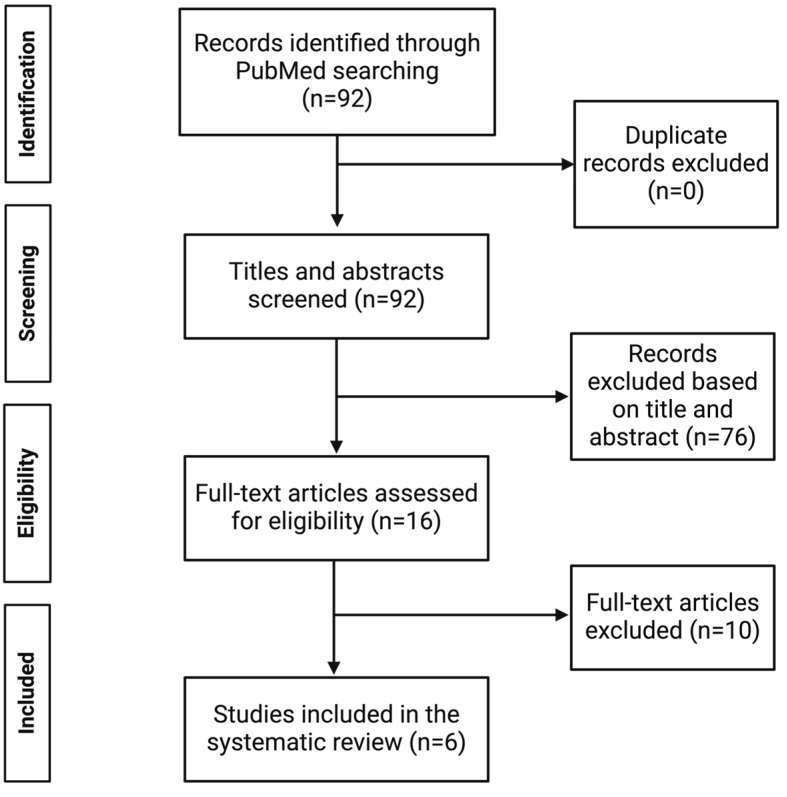
PRISMA flow diagram of the study selection process.

**Figure 3 toxins-16-00345-f003:**
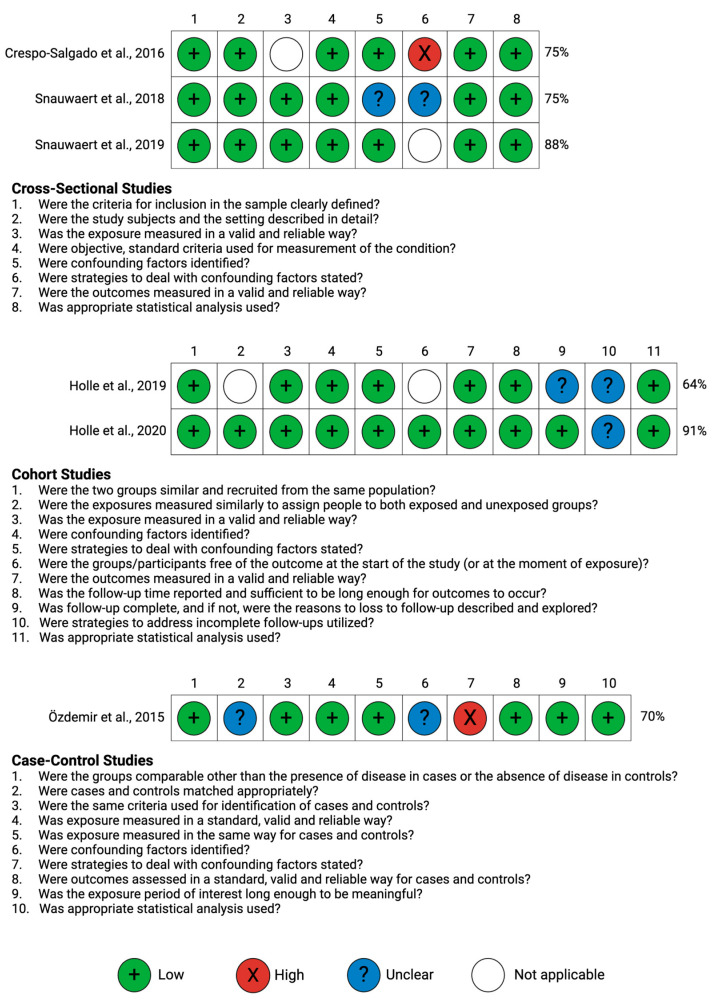
Risk of bias. Joanna Briggs Institute Critical Appraisal Checklist. Created with BioRender [19,20,21,22,23,24].

**Figure 4 toxins-16-00345-f004:**
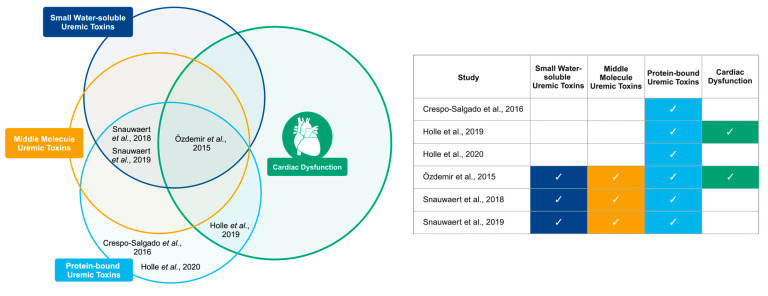
Venn diagram depicting the categorization of studies based on the classes of uremic toxins examined and their association with cardiac dysfunction. The diagram illustrates the overlap between studies investigating small water-soluble uremic toxins, middle molecule uremic toxins, and protein-bound uremic toxins. The adjacent table provides a summary of each study’s focus, indicating the specific uremic toxin classes examined and whether cardiac dysfunction was assessed. Created with BioRender [19,20,21,22,23,24].

**Table 1 toxins-16-00345-t001:** Study participant demographics of eligible articles.

Study Reference	Country	Patient Recruitment	Type of Study	Total Participants (n)	Gender Male	Age Mean (y)	Patient Group		Control Group, eGFR (mL/min/1.73 m^2^)
							Dialysis Status	eGFR (mL/min/1.73 m^2^)	
Crespo-Salgado et al., 2016 [19]	USA	June 2014 to March 2015	Cross-sectional study	39	64.1%	12.1	Established dialysis (HD and PD)	- HD: <15 - PD: <15	- Healthy children, 118 [85; 139]
Holle et al., 2019 [20]	12 European countries	October 2009 to August 2011	Prospective cohort study	609	65.6%	12.1	Pre-dialysis	- All: 28.1 ± 10.3 - CKD 3a: 51.4 ± 4.1 - CKD 3b: 36.0 ± 4.1 - CKD 4: 22.5 ± 4.1 - CKD 5: 13.3 ± 1.4	- Not specified
Holle et al., 2020 [21]	12 European countries	October 2009 to August 2011	Prospective cohort study	604	66%	12.1	Pre-dialysis	- 26.6 ± 11.2	- Not specified
Özdemir et al., 2015 [22]	Turkey	N/A	Case-control study	40	55%	12.0	Established dialysis (HD and PD)	- Not specified	- Healthy children
Snauwaert et al., 2018 [23]	Belgium	August 2014 to July 2016	Prospective cross-sectional study	107	67.5%	7.8	Pre-dialysis	- All: 48 [24; 71] - CKD 1–2: 74 [67; 103] - CKD 3: 47 [35; 55] - CKD 4: 20 [18; 26] - CKD 5: 11 [9; 13]	- Healthy children, 137 [119; 159]
Snauwaert et al., 2019 [24]	Belgium	2011 to 2017	Prospective cross-sectional study	194	36%	12.3	Pre-dialysis	- CKD 4–5: 17 [11; 23]	- Healthy children, 137 [119; 159]

Data are mean ± SD, median (IQR) or median [25th; 75th percentiles] as appropriate.

**Table 2 toxins-16-00345-t002:** A summary of studies discussing relationships between uremic toxin concentrations and cardiac structural and functional changes.

Study	Key Cardiac Findings	Methods Used	Quantitative Results	Recommendations	Implications for Cardiac Health
Holle et al., 2019 [20]	- IS significantly associated with cIMT SDS at baseline and with the progression of PWV SDS within 12 months.	- Serum analysis using HPLC and fluorescence detection - Ultrasound device for cIMT - Oscillometric Vicorder for PWV - Two-dimensional echocardiography for LV mass and LVMI	- Uremic toxin concentrations - CV parameters - Correlation coefficients - Regression coefficients	- Prospective evaluation of gut-derived uremic toxins and CV parameters. - Revaluate the impact of pre-emptive kidney transplants and other renal replacement strategies. - Important to monitor vascular changes in pediatric CKD patients.	- Higher IS levels may be associated with developing CVD through increased arterial stiffness.
Özdemir et al., 2015 [22]	- CV morbidity significantly associated with the clearance of middle and large molecular weight uremic toxins in children undergoing dialysis. - Patients on HD showed worse CV parameters, than those on PD.	- Blood tests to measure toxin clearance - Vicorder for AIx and PWV - Ultrasonography for cIMT - Echocardiography for LV systole and diastole measurements and LV mass	- Uremic toxin clearance - CV parameters	- Monitor the clearance of middle and large molecular weight uremic toxins in children receiving dialysis. - Assess PD’s efficiency for clearing middle- and large-molecular weight uremic toxins.	- HD may have limitations in removing protein-bound uremic toxins, suggesting PD may be more efficient.

Abbreviations: AIx: augmentation index; cIMT: carotid intima-media thickness; CKD: chronic kidney disease; CV: cardiovascular; CVD: cardiovascular disease; HD: hemodialysis; HPLC: high-performance liquid chromatography; IS: indoxyl sulfate; LV: left ventricular; LVMI: left ventricular mass index; PD: peritoneal dialysis; PWV: pulse wave velocity; SDS: standard deviation score.

## Data Availability

Data is contained within the article or Appendix A.

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
