# Peer review of "A Systematic Review of Uremic Toxin Concentrations and Cardiovascular Risk Markers in Pediatric Chronic Kidney Disease"

_toxins, 2024, doi:10.3390/toxins16080345_

Round 1

Reviewer 1 Report

Comments and Suggestions for Authors

SEE ATTACHED FILE

Thank you for the opportunity to review the manuscript (ID: toxins-3049267) titled “A Systematic Review of Uremic Toxin Concentrations and Cardiac Changes in Pediatric Cardiorenal Syndrome”. Please see my comments below:

1)      This is a narrative review and not a systematic review. The research question is defined either poorly or not at all. The quality of the 8 studies is not adequately assessed and one of them is actually a case report of n=1. There is no systemic review of the individual or collective data from these studies.

2)      There is a lot of important information missing in both the text and summary table of the studies presented. These were predominantly cohort studies in pre-dialysis CKD and some dialysis patients. The authors did not provide any quantitative values or statistical spread of uraemic toxin values in either patients or healthy controls. What is the normal reference range in healthy toxins and how do these values compare to those in CKD patients. The authors describe that severity of CKD appears to correlate with uraemic toxin levels but do not actually show us this relationship

3)       What was the eGFR of patients with pre-dialysis CKD? Which toxins are directly related to renal excretion versus other factors? How were toxins measured ie. what assays were used and were these samples collected pre- or post-HD sessions or on non-dialysis days?

4)      The authors mentioned a number of uraemic toxins including indoxyl sulfate, p-cresol, ADMA, IAA and HA. Why were these toxins chosen? What was the underlying rationale of the studies which chose to examine these toxins?

5)      The authors stated a specific interest in investigating the relationship between uraemic toxins and the cardiorenal syndrome. However, the cardiovascular parameters mentioned include carotid intimal-media thickness and non-invasive measurement of arterial stiffness. Only one study (which was actually a case report) discussed a patient with cardiac dysfunction which improved after HD. The authors concluded that this was due to clearance of uraemic toxins through HD and this improved further after kidney transplant. It is unclear how this conclusion was actually reached when there was no information on uraemic toxins before and after starting dialysis to actually show this? Ultimately, there doesn’t seem to be any meaningful data, discussion or analysis regarding cardiorenal syndrome and uraemic toxins.

6)      There was a quite tangential discussion point regarding differences in dialysis modality in terms of uraemic toxin clearance. The discussion section suggests that the authors believe that PD is superior in terms of middle molecule and protein bound clearance? Again, there is no conclusive evidence that either HD or PD is superior in terms of uraemic solute clearance? Furthermore, multiple studies have failed to find a consistent advantage of one dialysis modality over the other in terms of cardiovascular outcomes. This section would need to be removed entirely or at least heavily modified with in more depth data and meaningful scientific discussion.

7)      The abstract also does not summarise the manuscript!

Author Response

*All revisions are in red*

Comment 1: “"This is a narrative review and not a systematic review. The research question is defined either poorly or not at all. The quality of the 8 studies is not adequately assessed and one of them is actually a case report of n=1. There is no systemic review of the individual or collective data from these studies."

Response 1: Thank you for your valuable feedback. We agree that the initial manuscript lacked clarity in defining the research question and assessing the quality of the included studies. In the revised manuscript, we have ensured a clear and precise definition of the research question: "What is the relationship between uremic toxin levels and cardiac changes in pediatric patients with CKD?" [page 4, lines 146-152] Additionally, we have performed a quality assessment of each included study, excluding the case report to maintain the integrity of a systematic review [page 4, lines 169-171; page 5, lines 178-181]. 

Comment 2: "There is a lot of important information missing in both the text and summary table of the studies presented. These were predominantly cohort studies in pre-dialysis CKD and some dialysis patients. The authors did not provide any quantitative values or statistical spread of uraemic toxin values in either patients or healthy controls. What is the normal reference range in healthy toxins and how do these values compare to those in CKD patients? The authors describe that severity of CKD appears to correlate with uraemic toxin levels but do not actually show us this relationship."

Response 2: In response, we have included quantitative values of uremic toxin levels in both patients [supplementary materials, Table S2]. Table S2 also includes z-scores for uremic toxin levels in CKD patients compared with levels observed in healthy populations. We are unable to perform our own statistics (and any statistics that we perform would be inaccurate) due to the vast differences between studies in terms of population demographics, the assay types used, the different measurement units of toxins, etc. The relationship between CKD severity and uremic toxin levels is now demonstrated with reference to statistical analyses reported in the article and reference to visual aids in the study, as presented in the revised Results section and summary tables [page 8, lines 256-272].

Comment 3: "What was the eGFR of patients with pre-dialysis CKD? Which toxins are directly related to renal excretion versus other factors? How were toxins measured, i.e., what assays were used and were these samples collected pre- or post-HD sessions or on non-dialysis days?"

Response 3: We have now detailed the eGFR values of patients with pre- and post-dialysis CKD in the revised manuscript. Furthermore, we have specified which toxins are primarily excreted by the kidneys and have proposed other factors for those not influenced by renal excretion [pages 8-9, lines 272-281]. We have also elaborated on the methods used for toxin measurement, including the specific assays and the timing of sample collection relative to HD sessions, which can be found in the revised Methods section [Supplementary Materials, Table S2].

Comment 4: "The authors mentioned a number of uraemic toxins including indoxyl sulfate, p-cresol, ADMA, IAA and HA. Why were these toxins chosen? What was the underlying rationale of the studies which chose to examine these toxins?"

Response 4: We acknowledge the need for clarity on the selection of uremic toxins. In the revised manuscript, we have provided a rationale for the selection of specific uremic toxins, explaining their known or suspected roles in cardiovascular pathology and their prevalence in the existing pediatric CKD literature. This information is now included in the Introduction section [page 3, lines 106-125].

Comment 5: "The authors stated a specific interest in investigating the relationship between uraemic toxins and the cardiorenal syndrome. However, the cardiovascular parameters mentioned include carotid intimal-media thickness and non-invasive measurement of arterial stiffness. Only one study (which was actually a case report) discussed a patient with cardiac dysfunction which improved after HD. The authors concluded that this was due to clearance of uraemic toxins through HD and this improved further after kidney transplant. It is unclear how this conclusion was actually reached when there was no information on uraemic toxins before and after starting dialysis to actually show this? Ultimately, there doesn’t seem to be any meaningful data, discussion or analysis regarding cardiorenal syndrome and uraemic toxins."

Response 5: In the revised manuscript, we have strengthened the discussion on the relationship between uremic toxins and cardiorenal syndrome by incorporating data from studies. We have included additional information on cardiovascular parameters, provided more detailed and critical analyses, and clarified how conclusions were reached [page 10, lines 343-349; page 10, table 2]. This enhances the linkage between uremic toxin clearance and improvements in cardiovascular function.

Comment 6: "There was a quite tangential discussion point regarding differences in dialysis modality in terms of uraemic toxin clearance. The discussion section suggests that the authors believe that PD is superior in terms of middle molecule and protein-bound clearance? Again, there is no conclusive evidence that either HD or PD is superior in terms of uraemic solute clearance? Furthermore, multiple studies have failed to find a consistent advantage of one dialysis modality over the other in terms of cardiovascular outcomes. This section would need to be removed entirely or at least heavily modified with in-depth data and meaningful scientific discussion."

Response 6: We agree and have significantly revised the discussion on dialysis modalities in the revised manuscript. To reflect this change, we have included a dedicated section titled “2.3 Distinctions between dialysis and non-dialysis patients” in the Results comparing patients with CKD on dialysis versus those not on dialysis, highlighting the significant differences in uremic toxin clearance, cardiovascular outcomes, and the role of residual kidney function [page 9, lines 317-323; page 12, lines 373-417]. This addition addresses the major discriminatory factors in clinical practice and enhances the manuscript's relevance.

Comment 7: "The abstract also does not summarize the manuscript!"

Response 7: We have rewritten the abstract [page 1, lines 5-6, 9-11, 16-17, 20-26] to provide a concise and accurate summary of the manuscript, clearly outlining the research question, methodology, key findings, and implications.

We believe that these revisions address all your concerns and significantly enhance the clarity and utility of our manuscript. Thank you for your constructive feedback, which has been instrumental in improving the quality of our work.

Reviewer 2 Report

Comments and Suggestions for Authors

Unfortunately, I have to vote to reject this manuscript in its current version

I hoped the manuscript would be a reference for readers of Cardiorenal syndrome (CRS) in pediatric patients. I was not satisfied. The introduction and Figure 1 are at minimum basic level knowledge. 

The current manuscript is a summary of 7 articles with one case report. The other angles of "cardiorenal syndrome" such as acute CRS and the effect of cardiovascular disease on kidneys are not mentioned. It does not distinguish between patients with CKD on dialysis or off dialysis. Major discriminatory factors in clinical practice. 

Looking at Figure 3. the "Venn diagram depicts" only 3 articles are placed inside the "cardiac dysfunction circle" with one case report.

 I don't think this manuscript brings any extra information more than the summary of each individual articles.

Comments on the Quality of English Language

I don't see a major issue with the quality of the English language. 

Author Response

Comment 1: “Unfortunately, I have to vote to reject this manuscript in its current version
I hoped the manuscript would be a reference for readers of Cardiorenal syndrome (CRS) in pediatric patients. I was not satisfied. The introduction and Figure 1 are at minimum basic level knowledge.”

Response 1: Thank you very much for your feedback! We have significantly revised the introduction to include a more detailed overview of CRS and its pathophysiology, subtypes of CRS [page 1, lines 33-43], and classes of uremic toxins [page 3, lines 80-93], with a focus on pediatric patients. The revised version also addresses the unique mechanisms and clinical implications of CRS in this population [page 2, lines 52-59], providing a more comprehensive background that will serve as a valuable reference for readers. Additionally, we have completely redone Figure 1 [page 2, line 60] to illustrate higher-level concepts from the different origins of uremic toxins to their various destinations. We incorporated text into the revised Figure 1 to make it more descriptive [page 2, lines 66-72].

Comment 2: “The current manuscript is a summary of 7 articles with one case report. The other angles of "cardiorenal syndrome" such as acute CRS and the effect of cardiovascular disease on kidneys are not mentioned. It does not distinguish between patients with CKD on dialysis or off dialysis. Major discriminatory factors in clinical practice.”

Response 2: We appreciate your comments regarding the scope of the manuscript. In response, we included the different types of CRS, including acute CRS, and mentioned the bidirectional nature of CRS [page 1, lines 33-43; page 2, line 60; page 2, lines 66-72]. Our review primarily focuses on CRS type 4 (chronic kidney disease resulting in chronic heart failure), which we now clarify in the inclusion/ exclusion criteria under Methods [page 14, lines 499-508 and lines 515-518]. Additionally, we have included a dedicated section in the Results comparing patients with CKD on dialysis versus those not on dialysis, highlighting the significant differences in uremic toxin clearance, cardiovascular outcomes, and the role of residual kidney function [page 10, lines 292-336]. This addition addresses the major discriminatory factors in clinical practice and enhances the manuscript's relevance.

Comment 3: “Looking at Figure 3. the "Venn diagram depicts" only 3 articles are placed inside the "cardiac dysfunction circle" with one case report.”

Response 3: We have revised Figure 3 to enhance clarity and provide a more summarized overview of the studies included [page 6, lines 159-165]. The Venn diagram, and now the accompanying tabulated chart depicts whether each study addressed specific groups of uremic toxins and cardiac dysfunction. Additionally, we have also included three new tables (Table 2 [page 10, line 284] and Tables S1 [page 15, line 535 + in supplemental materials] and S2 [page 21, line 562 + in supplemental materials]) that provide a more detailed and comprehensive summary of the reviewed literature. These tables include the nephro-cardiac interplay and main findings in terms of uremic toxin changes and cardiac changes, offering a concise, clear and more informative presentation of the text in the main body.

Comment 4: “I don't think this manuscript brings any extra information more than the summary of each individual article.”

Response 4: In response to your concern, we have critically analyzed the findings from the reviewed articles and have re-synthesized this information to provide new insights regarding CRS type 4 in pediatric patients [page 11, lines 344-350; page 11, lines 363-366; page 12, lines 411-427]. Our discussion section now includes a detailed analysis of the implications of these findings for clinical practice and future research [page 12-13, lines 440-457]. In terms of future directions, we also propose new hypotheses and therapeutic approaches based on the synthesized data [page 13, lines 458-476], thereby adding substantial value beyond a summary of individual articles.

We believe these extensive revisions address your concerns and significantly enhance the manuscript's contribution to the field. We appreciate your constructive feedback, which has been instrumental in improving the quality of our work. Thank you for your consideration.

Reviewer 3 Report

Comments and Suggestions for Authors

The paper entitled: “A Systematic Review of Uremic Toxin Concentrations and Car-2 diac Changes in Pediatric Cardiorenal Syndrome” focusses on the role of uremic toxins in pediatric cardiorenal syndrome. The available literature is limited and the information provided, is laying the groundwork for tailored treatments to enhance clinical outcomes and quality of life for affected children.

Specific comments

Abstract

-“It is postulated that the accumulation of uremic toxins in the blood stream, as a consequence of declining kidney function, may be responsible for these adverse cardiac effects.” Instead of “responsible for” please write “contribute to”

-A brief conclusion in the abstract would be needed

Introduction

Figure 1: an arrow going from the heart to the kidney is missing. Also in the legend to the figure the heart is not mentioned. The definition of uremic toxin can be refined referring to their biological activity. In addition, uremic toxins do not only originate from gut bacterial metabolism but also from the diet and mammalian metabolism.

Uremic toxin is not a synonym for uremic retention solutes (line 49). All metabolites that accumulate in the circulations with a decreased in kidney function are called uremic retention solutes, however before they can be called uremic toxins, biological activity needs to be demonstrated in in vitro or in vivo experimental settings.

Line 64: “pediatric patients with CKD” instead of “pediatric CKDs”

Line 77: Could the authors be more specific on the exclusion criteria for the selected papers

Table 1: it would be of interest to add outcome parameters of major aim of the study in the table

Minor

Replace “renal” by “kidney” line 93 end-stage kidney disease (ESKD)

Comments on the Quality of English Language

-

Author Response

Comment 1: ““It is postulated that the accumulation of uremic toxins in the bloodstream, as a consequence of declining kidney function, may be responsible for these adverse cardiac effects.” Instead of “responsible for” please write “contribute to””

Response 1: Thank you for your feedback. We have revised the sentence in the abstract to read: "It is postulated that the accumulation of uremic toxins in the bloodstream, as a consequence of declining kidney function, may contribute to these adverse cardiac effects." [page 1, lines 4-7]

Comment 2: “A brief conclusion in the abstract would be needed”

Response 2: A brief conclusion has been added to the abstract to summarize the key findings and implications of the study [page 1, lines 16-19].

Comment 3: “Figure 1: an arrow going from the heart to the kidney is missing. Also in the legend to the figure the heart is not mentioned. The definition of uremic toxin can be refined referring to their biological activity. In addition, uremic toxins do not only originate from gut bacterial metabolism but also from the diet and mammalian metabolism.”

Response 3: We have completely redone Figure 1 [page 2, lines 60-72] to illustrate higher-level concepts from the different origins of uremic toxins to their various destinations. We incorporated text into the revised Figure 1 to make it more descriptive. We have added an arrow from the heart to the kidney and included the heart in the figure legend [page 2 for figure]. The definition of uremic toxin has been refined to refer to their biological activity [page 3, lines 75-77; page 3, lines 80-96], and we have clarified that uremic toxins can originate from gut bacterial metabolism, diet, and mammalian metabolism (in both Figure 1 and in the text) [page 2 for figure; page 3, lines 77-79].

Comment 4: “Uremic toxin is not a synonym for uremic retention solutes (line 49). All metabolites that accumulate in the circulation with a decrease in kidney function are called uremic retention solutes; however, before they can be called uremic toxins, biological activity needs to be demonstrated in in vitro or in vivo experimental settings.”

Response 4: We have replaced the mention of “uremic retention solutes” with “uremic toxins” [page 3, line 74]and specified the distinction between the two terms in the text [page 3, lines 96-102].

Comment 5: “Line 64: “pediatric patients with CKD” instead of “pediatric CKDs””

Response 5: We have corrected the phrase to "pediatric patients with CKD." [page 3, line 117]

Comment 6: “Line 77: Could the authors be more specific on the exclusion criteria for the selected papers”

Response 6: We have revised the methods section to provide more specific details on the exclusion criteria for the eligible articles, including the exclusion of non-human studies, research involving adult subjects, articles on CRS types 1, 2, 3, and 5, studies where the primary disease was acute kidney injury, and studies in which cardiac dysfunction preceded renal impairment in the sequence of organ involvement [page 14, lines 499-508].

Comment 7: “Table 1: it would be of interest to add outcome parameters of major aim of the study in the table”

Response 7: We have created a new table to include outcome parameters that reflect the major aims of the studies reviewed. Since Table 1 focuses on participant demographics, we included Table S1 [page 15, line 535 + in supplemental materials], which discusses the aims, outcome parameters, nephro-cardiac interplay, uremic toxin changes, and cardiac changes of each study (where applicable). This addition provides a more concise and clearer understanding of the study outcomes and their relevance to our review.

Comment 8: “Replace “renal” by “kidney” line 93 end-stage kidney disease (ESKD)”

Response 8: The term "renal" has been replaced with "kidney" on line 93 to read "end-stage kidney disease (ESKD)." [ape 6, line 167]

We believe these extensive revisions address your concerns and significantly enhance the manuscript's contribution to the field. We appreciate your constructive feedback, which has been instrumental in improving the quality of our work. Thank you for your consideration.

Reviewer 4 Report

Comments and Suggestions for Authors

A review of the manuscript titled „A Systematic Review of Uremic Toxin Concentrations and Car-2 diac Changes in Pediatric Cardiorenal Syndrome” submitted to Toxins.

The manuscript presents a systematic review of the literature on the relationship between uremic toxin levels in chronic kidney disease (CKD), their effects on cardiac parameters, and the resulting cardiac dysfunction in children. This review follows PRISMA guidelines and is based on a systematic search using the terms "uremic toxins" and "cardiorenal syndrome." The review includes full-text articles written in English, covering subjects from birth to 18 years of age, and published through December 2021. The systematic review is registered on PROSPERO.

My comments:

1. Lack of Connection Between Nephrological and Cardiac Disorders:

Comment: In the presented systematic review, there is no clear connection between nephrological and cardiac disorders in children with cardiorenal syndrome. Is it true that the authors of the analyzed works do not include such analyses, comparisons, or correlations?

Recommendation: The authors should clarify whether the reviewed studies addressed the interplay between nephrological and cardiac disorders in pediatric cardiorenal syndrome. If such analyses were included in the original works but not highlighted in the review, it would be beneficial to summarize these findings explicitly.

2. Presentation of Relationship Data:

Comment: Including the information from the section "Relationship between uremic toxin concentrations and cardiac structural and functional changes" in a table might improve clarity.

Recommendation: Presenting this information in a table could enhance the clarity and accessibility of the data.

3. Inclusion of Summary Tables or Figures:

Comment: It would be desirable to include tables or figures summarizing changes occurring in plasma/serum and urine, as well as those detected in imaging tests, such as echocardiography.

Recommendation: Adding tables or figures to summarize the biochemical changes in plasma/serum and urine, along with imaging findings, would provide a comprehensive overview of the systemic and cardiac alterations observed in pediatric CRS. This visual representation could aid in better understanding the extent and impact of uremic toxins on cardiac health.It would be desirable to include tables or figures summarizing changes occurring in plasma/serum, and urine, as well as those detected in imaging tests, e.g. echocardiography.

Author Response

Comment 1: 

“Comment: In the presented systematic review, there is no clear connection between nephrological and cardiac disorders in children with cardiorenal syndrome. Is it true that the authors of the analyzed works do not include such analyses, comparisons, or correlations? 

Recommendation: The authors should clarify whether the reviewed studies addressed the interplay between nephrological and cardiac disorders in pediatric cardiorenal syndrome. If such analyses were included in the original works but not highlighted in the review, it would be beneficial to summarize these findings explicitly.”

Response 1: Thank you very much for your feedback! In response, we have thoroughly reviewed the analyzed the eligible studies to clarify whether they addressed the interplay between nephrological and cardiac disorders in pediatric CRS. To address this, we have revised the manuscript to summarize these findings into a new table, Table S1 [page 15, line 535 + in supplemental materials], that highlights the connections between nephrological dysfunction and cardiac complications, emphasizing the mechanisms and outcomes observed in the reviewed studies. This table provides a clear and concise overview of the interplay between nephrological and cardiac disorders in pediatric CRS.

Comment 2: 

“Comment: Including the information from the section "Relationship between uremic toxin concentrations and cardiac structural and functional changes" in a table might improve clarity.

Recommendation: Presenting this information in a table could enhance the clarity and accessibility of the data.”

Response 2: We have created a new table, Table 2 [page 8, line 284], that presents the relationship between uremic toxin concentrations and cardiac structural and functional changes. This table summarizes the key findings from the relevant reviewed studies, the methods used, quantitative results, recommendations and implications for cardiac health, providing a clear and concise overview of the data. 

Comment 3: 

“Comment: It would be desirable to include tables or figures summarizing changes occurring in plasma/serum and urine, as well as those detected in imaging tests, such as echocardiography.

Recommendation: Adding tables or figures to summarize the biochemical changes in plasma/serum and urine, along with imaging findings, would provide a comprehensive overview of the systemic and cardiac alterations observed in pediatric CRS. This visual representation could aid in better understanding the extent and impact of uremic toxins on cardiac health.”

Response 3: 

We have added another new table, Table S2 [page 21, line 562 + in supplemental materials], to summarize the uremic toxin concentration changes in plasma/serum, as well as imaging findings from echocardiography and other relevant tests as per the eight eligible studies. 

We believe these extensive revisions address your concerns and significantly enhance the manuscript's contribution to the field. We appreciate your constructive feedback, which has been instrumental in improving the quality of our work. Thank you for your consideration.

Round 2

Reviewer 1 Report

Comments and Suggestions for Authors

Thank you for the opportunity to review the manuscript revisions (ID: toxins-3049267) titled “A Systematic Review of Uremic Toxin Concentrations and Cardiac Changes in Pediatric Cardiorenal Syndrome”. Please see my comments below:

1)      The tables (both within the manuscript and those found in supplementary material) remain very poorly organised. I found this very difficult to read and understand. Salient specific points would include

·         Table 1 tries to summarise the basic characteristics of each study. However, are these in pre-dialysis CKD or established dialysis patients? This information should be included for this table specifically. Levels of GFR (for the predialysis patients and healthy control cohorts) should also be included here. I note the authors (in response 3) said that they have ‘now detailed eGFR values” in the manuscript. However, I was unable to find any such information with the exception of the study by Snauwaert which stated “that GFR values in CKD stages 1-2 were 44, 42 and 39 according to 3 different equations”. Is this an incorrect typo? The KDIGO CKD criteria clearly states that eGFR 3 is <60mL/min/1.73m2. Furthermore, there was no information on levels of GFR reported in any of the other studies included?

·         Both supplementary tables are almost impossible to read or understand. Can the authors present these information in landscape rather than portrait format? Also: what were the values of uraemic toxin levels in healthy paediatric patients? Did these patients have normal GFR? The authors did mention this in their discussion section in the manuscript but I was unable to find what the actual values were.

2)      The inclusion of a case report of n=1 is questionable. How could this be considered as ‘high quality’ robust evidence? The conclusion that the initiation of HD and restoration of kidney function through transplantation led to improved cardiac function as a result of uraemic toxin clearance is also debatable especially when there were no pre or post measurements performed. Could improvements in fluid balance or blood pressure not also have improved the cardiac function in this patient? Additionally, while there is no data (to my knowledge) of increased dialysis dose/intensity in paediatric populations, there have been 2 randomised controlled trials in adult populations (both published in NEJM) demonstrating that daily frequent/nocturnal dialysis MAY improve surrogate CVD measurements such as LV mass. However, more frequent dialysis did NOT improve mortality in adult dialysis patients and there have not been subsequent post-hoc analyses on whether there were favourable changes in cardiac function (to my knowledge). It would be inappropriate to include this case report from my perspective.

3)      In page 8, there is discussion about a higher burden of CVD in HD compared to PD patients based on increased incidence of LVH. As per my previous comments to the authors, there remains no conclusive evidence that there is a more favourable CVD risk profile associated with any particular dialysis modality. To then postulate that this could be due to differences in uraemic solute clearance is not scientifically correct: “HD patients with CVD had significantly lower clearance rates of B2M and homocysteine compared to HD patients without CVD”. All the data presented in this review indicates that the main uraemic toxins implicated in CVD are p-cresol and indoxyl sulfate. Are the authors then assuming that because homocysteine and B2M clearance is lower that this also extrapolates to protein bound uraemic solute removal? In addition, there is a final statement which says “PD patients did not show significant differences in uraemic toxin levels between those with and without CVD, indicating more efficient toxin clearance with PD”. How does this statement make sense?

4)      A systemic review should include some sort of data summation and/or forest plot which demonstrates the strength of the association between uraemic toxins and CVD risk markers. There is nothing available with regards to this? Otherwise this should not be titled as a systematic review.

5)      Were there any negative studies which failed to demonstrate an association between uraemic toxins and CVD markers in paediatric CKD patients?

6)      Outside of the case report (n=1) there are no studies which actually look at cardiac function. The surrogate CVD risk markers included in the published studies are focus upon LV mass, LVH, CIMT or arterial stiffness. Do the authors want to revise their title? There doesn’t seem to be any evidence specifically with regards to cardiorenal syndrome in this manuscript.

Author Response

Thank you for your valuable feedback and insightful comments on our manuscript. We appreciate your thorough review and are grateful for the opportunity to improve our work based on your recommendations.

Comment 1: “Table 1 tries to summarise the basic characteristics of each study. However, are these in pre-dialysis CKD or established dialysis patients? This information should be included for this table specifically. Levels of GFR (for the predialysis patients and healthy control cohorts) should also be included here.”

Response 1: We have updated Table 1 [page 7] and added a new column titled “Patient Group” to specify whether the patients in each study are pre-dialysis CKD or established dialysis patients. Additionally, we have included detailed eGFR levels for the pre-dialysis patients and healthy control cohorts, as specified in the respective studies. We have also incorporated this information into the manuscript text, particularly in the Results section, to provide a comprehensive overview of the patient populations and the severity of CKD [pages 9-10, lines 194-231].

Comment 2: “I note the authors (in response 3) said that they have ‘now detailed eGFR values” in the manuscript. However, I was unable to find any such information with the exception of the study by Snauwaert which stated “that GFR values in CKD stages 1-2 were 44, 42 and 39 according to 3 different equations”. Is this an incorrect typo? The KDIGO CKD criteria clearly states that eGFR 3 is <60mL/min/1.73m2.”

Response 2: The highlighted eGFR values were an error and have been updated with the accurate eGFR values from the paper [page 11, lines 317-320]. We acknowledge the KDIGO CKD criteria and have ensured that the classifications align with these standards.

Comment 3: “Furthermore, there was no information on levels of GFR reported in any of the other studies included?”

Response 3: We added a new subsection under the Results section titled “2.1 Patient characteristics and eGFR levels” [pages 9-10, lines 194-231] where we included all specified eGFR levels in the eligible 8 studies. This information was only found in some studies, as eGFR values were of limited relevance to particular studies.

Comment 4: “Both supplementary tables are almost impossible to read or understand. Can the authors present these information in landscape rather than portrait format?”

Response 4: The tables have been reformatted to landscape format [Supplementary Materials, pages 22-21]. It is challenging to format the tables into a more readable layout due to the large amount of content in each table, but we have made these adjustments as per the requests of other reviewers.

Comment 5: “Also: what were the values of uraemic toxin levels in healthy paediatric patients? Did these patients have normal GFR? The authors did mention this in their discussion section in the manuscript but I was unable to find what the actual values were.”

Response 5: In table 1 [page 7], we added a new column titled “Control Group” where we have provided specific details about the control groups in each study, including the eGFR values where available.

Comment 6: “The inclusion of a case report of n=1 is questionable. How could this be considered as ‘high quality’ robust evidence? The conclusion that the initiation of HD and restoration of kidney function through transplantation led to improved cardiac function as a result of uraemic toxin clearance is also debatable especially when there were no pre or post measurements performed. Could improvements in fluid balance or blood pressure not also have improved the cardiac function in this patient? Additionally, while there is no data (to my knowledge) of increased dialysis dose/intensity in paediatric populations, there have been 2 randomised controlled trials in adult populations (both published in NEJM) demonstrating that daily frequent/nocturnal dialysis MAY improve surrogate CVD measurements such as LV mass. However, more frequent dialysis did NOT improve mortality in adult dialysis patients and there have not been subsequent post- hoc analyses on whether there were favourable changes in cardiac function (to my knowledge). It would be inappropriate to include this case report from my perspective.”

Response 6: We appreciate the reviewer’s concerns regarding the inclusion of the case report. The case report by Nehus et al. (2011) meets our inclusion criteria (as detailed in the Methods section, pages 20-21, lines 685-694). Furthermore, the inclusion of this case report also provides relevant clinical observations on the impact of intensive HD and transplantation on cardiac function in a pediatric patient with CKD. While we acknowledge that case reports do not provide high-quality evidence comparable to randomized controlled trials due to the small sample size, the findings of the authors are noteworthy as they offer valuable clinical insights, especially in an area such as ours where research is scarce.

The pediatric CKD population is underrepresented in the literature, and the limited number of studies necessitates the inclusion of all relevant studies that are eligible to illustrate clinical outcomes and generate hypotheses for future research. Although improvements in fluid balance and blood pressure likely contributed to the observed cardiac improvements, excluding such reports would narrow the scope of our review and overlook critical clinical experiences.

Regarding the evidence from adult populations, we recognize that randomized controlled trials have shown that increased dialysis dose/intensity can improve surrogate CVD measurements, such as left ventricular mass, without necessarily impacting mortality. However, it would be inaccurate and complex to translate these findings to pediatric patients (as explained in detail in the Introduction section of the manuscript on the physiological and pathological differences between adults and children) and highlight the need for more dedicated research in the pediatric population.

We have clearly stated the limitations of this evidence in our discussion to ensure that readers can appropriately interpret the findings. Including this case report provides a more comprehensive overview of the current state of knowledge and highlights areas needing further investigation, which we discuss in the manuscript.

Comment 7: “In page 8, there is discussion about a higher burden of CVD in HD compared to PD patients based on increased incidence of LVH. As per my previous comments to the authors, there remains no conclusive evidence that there is a more favourable CVD risk profile associated with any particular dialysis modality.”

Response 7: We appreciate the reviewer’s feedback regarding the comparison of CVD burden between HD and PD patients. Based on your comments, we have revised the manuscript to clarify that while some studies suggest a higher incidence of LVH among HD patients compared to PD patients, this does not necessarily imply a superior CVD risk profile for any specific dialysis modality. We acknowledge the potential for confounding factors that were not controlled for in the studies referenced [page 16, lines 454-457].

Comment 8: “To then postulate that this could be due to differences in uraemic solute clearance is not scientifically correct: “HD patients with CVD had significantly lower clearance rates of B2M and homocysteine compared to HD patients without CVD”. All the data presented in this review indicates that the main uraemic toxins implicated in CVD are p-cresol and indoxyl sulfate. Are the authors then assuming that because homocysteine and B2M clearance is lower that this also extrapolates to protein bound uraemic solute removal?”

Response 8: We have addressed the inaccuracy noted by the reviewer regarding the postulation that differences in uraemic solute clearance could explain the higher burden of CVD in HD patients. Our revised text now clearly states that improved clearance of specific toxins such as β2M and homocysteine does not necessarily translate to improved CVD outcomes, nor does it imply comprehensive clearance of all uremic toxins in their respective classes [page 16, lines 462-468].

Comment 9: “In addition, there is a final statement which says “PD patients did not show significant differences in uraemic toxin levels between those with and without CVD, indicating more efficient toxin clearance with PD”. How does this statement make sense?"

Response 9: In response to the reviewer's comment on the statement regarding PD patients not showing significant differences in uremic toxin levels between those with and without CVD, we have expanded our discussion to provide more context and clarification. We now emphasize the limitations of the available data and the need for careful interpretation. We have also included a broader discussion on the complexities and limitations of current evidence regarding the impact of dialysis modalities on CVD risk, acknowledging the need for further research to provide more definitive insights [pages 16-17, lines 470-491].

Comment 10: A systemic review should include some sort of data summation and/or forest plot which demonstrates the strength of the association between uraemic toxins and CVD risk markers. There is nothing available with regards to this? Otherwise this should not be titled as a systematic review.

Response 10: We appreciate the reviewer's comment regarding the inclusion of data summation and/or a forest plot. However, we would like to clarify that our manuscript is a systematic review rather than a systematic review and meta-analysis. As such, it is not mandatory to include a forest plot unless statistical pooling of data through meta-analysis is performed.

Moreover, we found that there is insufficient data available in the included studies to conduct a robust meta-analysis. Furthermore, based on our understanding, a meta-analysis should be performed when the included studies are of similar design and similar outcome measures; however, the studies included in our systematic review are extremely heterogenous in nature due to the scarcity of research. The existing studies on the association between uremic toxins and cardiovascular disease (CVD) risk markers in the pediatric population are limited, both in number and in the level of detail provided. This paucity of data precludes us from performing statistical analyses with sufficient power and reliability.

To address the reviewer's concern, we have provided a detailed data summation table within the manuscript, summarizing the key findings of the included studies and mentioning any associations regarding specific uremic toxin changes and CVD risk marker changes (Table S1; Supplementary Materials, pages 22-26).

Comment 11: “Were there any negative studies which failed to demonstrate an association between uraemic toxins and CVD markers in paediatric CKD patients?”

Response 11: Following a thorough literature search, we were unable to identify any studies that explicitly failed to show associations between uremic toxins and CVD markers in this patient population. We want to highlight that research surrounding pediatric cardiorenal syndrome is still very limited. The lack of extensive research regarding pediatric cardiorenal syndrome, as mentioned in the Introduction and Discussion sections of our manuscript, could contribute to the absence of studies reporting negative results.

Comment 12: “Outside of the case report (n=1) there are no studies which actually look at cardiac function. The surrogate CVD risk markers included in the published studies are focus upon LV mass, LVH, CIMT or arterial stiffness. Do the authors want to revise their title? There doesn’t seem to be any evidence specifically with regards to cardiorenal syndrome in this manuscript.”

Response 12: To better reflect the content and focus of our review, we have revised the title of our manuscript. The new title is: "A Systematic Review of Uremic Toxin Concentrations and Cardiovascular Risk Markers in Pediatric Chronic Kidney Disease" [page 1].

We believe this revised title more accurately represents the scope of our review, which primarily examines associations between uremic toxin concentrations and cardiovascular risk markers such as LV mass, LVH, CIMT, and arterial stiffness in pediatric CKD patients.

We are confident that the revisions made in response to your suggestions have significantly enhanced the quality of our manuscript. We appreciate your time and effort in reviewing our work and look forward to your favourable consideration.